# Multidrug-Resistant and Extensively Drug-Resistant *Escherichia coli* in Sewage in Kuwait: Their Implications

**DOI:** 10.3390/microorganisms11102610

**Published:** 2023-10-23

**Authors:** Mahdi A. Redha, Noura Al Sweih, M. John Albert

**Affiliations:** Department of Microbiology, College of Medicine, Kuwait University, Jabriya 46300, Kuwait; mahdi-ali-q8@hotmail.com (M.A.R.); nourah.alsuwaih@ku.edu.kw (N.A.S.)

**Keywords:** sewage, *Escherichia coli*, multidrug resistance, public health, global clones, Kuwait

## Abstract

In Kuwait, some sewage is discharged into the sea untreated, causing a health risk. Previously, we investigated the presence of pathogenic *E. coli* among the 140 isolates of *E. coli* cultured from the raw sewage from three sites in Kuwait. The aim of the current study was to characterize the antimicrobial resistance of these isolates and the implications of resistance. Susceptibility to 15 antibiotic classes was tested. Selected genes mediating resistance to cephalosporins and carbapenems were sought. ESBL and carbapenemase production were also determined. Two virulent global clones, ST131 and ST648, were sought. A total of 136 (97.1%), 14 (10.0%), 128 (91.4%), and 2 (1.4%) isolates were cephalosporin-resistant, carbapenem-resistant, multidrug-resistant (MDR), and extensively drug-resistant (XDR), respectively. Among the cephalosporin-resistant isolates, *ampC*, *bla*_TEM_, *bla*_CTX-M_, *bla*_OXA-1_, and *bla*_CMY-2_ were found. Eighteen (12.9%) samples were ESBL producers. All carbapenem-resistant isolates were negative for carbapenemase genes (*bla*_OXA-48_, *bla*_IMP_, *bla*_GES_, *bla*_VIM_, *bla*_NDM_, and *bla*_KPC_), and for carbapenemase production. Resistance rates in carbapenem-resistant isolates to many other antibiotics were significantly higher than in susceptible isolates. A total of four ST131 and ST648 isolates were detected. The presence of MDR and XDR *E. coli* and global clones in sewage poses a threat in treating *E. coli* infections.

## 1. Background

Sewage undergoes treatment to reduce microbial contamination before discharge into the environment [1,2]. In Kuwait, 75% of sewage is treated, the treated effluent is used for landscaping and irrigation purposes, and the sludge after primary treatment is used as manure for growing plants and to remediate the soil. The remaining 25% of the sewage is discharged into the sea untreated, resulting in coastal water contamination [3,4]. The coastal sea is used for recreational purposes (swimming, boating, and fishing) in Kuwait, and sewage disposal takes place within a kilometer of recreational areas. Although treatment of sewage can greatly reduce microbial content, the treated effluent is still not fully safe and can be a threat to human health as it can harbor pathogenic and drug-resistant bacteria [1,2,5].

*Escherichia coli*, a fecal indicator, is a predominant organism of the sewage [6]. It is a causative agent of both intestinal and extra-intestinal infections due to diarrheagenic *E. coli* (DEC) and extra-intestinal pathogenic *E. coli* (ExPEC), respectively [7]. *E. coli* is also used as a sentinel organism for the surveillance of antimicrobial resistance (AMR) [8]. Cephalosporin and carbapenem antibiotics are important therapeutic agents for the treatment of *E. coli* infections [9,10]. *E. coli* clones, such as the sequence types, ST131 and ST648, are multi-drug resistant and highly virulent and have spread globally causing urinary tract infections and bloodstream infections [11]. These clones belong to certain Clermont phylogenetic groups [12,13]. In a previous study [14], we published our findings on 140 *E. coli* isolates cultured from raw sewage in Kuwait for DEC, ExPEC, and phylogenetic groups. In the present report, we present additional information on these *E. coli* isolates about resistance to a range of antimicrobial agents, including selected genes encoding extended-spectrum β-lactamases (ESBLs) and carbapenemases. In addition, even though there are numerous global *E. coli* clones, we sought two of them, ST131 and ST648, because of their worldwide distribution, encompassing North America, Europe, Asia, and Africa, and their ease of detection via PCR assays [15]. This study has provided insights into the problem of drug resistance in *E. coli* in the community emanating from sewage. The previous study focused on virulence properties, and the current study focuses on drug resistance including that in selected global clones. The potential implications of human exposure to multidrug-resistant *E. coli* in sewage, for the treatment of infections, are also discussed in this manuscript.

## 2. Methods

*E. coli* culture. *E. coli* isolates originated from raw sewage samples. Raw sewage was sampled once a month for 12 months from three sites in Kuwait during May 2018–April 2019. A total of 140 different *E. coli* isolates were cultured from 36 sewage samples. The sampling method and characterization of *E. coli*, including DEC, ExPEC, and Clermont phylogenetic groups, have been described previously [14]. The isolates stocked in tryptic soy broth (Oxoid, Basingstoke, Hampshire, UK) with 15% glycerol at −80 °C were subcultured onto MacConkey agar (Oxoid). Single colonies, after reconfirmation as *E. coli* in the API-20E test (bioMerieux, 69280 Marcy l’Etoile, France), were used for the current study. There were 3 isolates of DEC (2 atypical enteropathogenic *E. coli* (aEPEC) isolates, J28 and H164, and 1 enterotoxigenic *E. coli* (ETEC) isolate, H51), and 14 isolates of ExPEC [14]. The remainder of the isolates were neither diarrheagenic pathogens nor extra-intestinal pathogens and, therefore, can be considered non-pathogenic.

The disk diffusion test was conducted following the CLSI guidelines [16] against 28 antibiotics belonging to 15 classes. Susceptibilities to polymyxins were tested by growth inhibition in a broth containing the antibiotic (see later test details) (Appendix A). *E. coli* colonies grown on brain–heart infusion agar (Oxoid) at 37 °C for 20 h were emulsified in saline to obtain a 0.5 MacFarland turbidity standard comparable to the density of a bacterial suspension with 5 × 10^5^ cfu/mL. This suspension was used to inoculate Mueller–Hinton agar (Oxoid), which was later overlaid with antibiotic discs. The plate was incubated at 37 °C for 18 h, and the growth inhibition zones were measured. Quality control organisms for the disc diffusion test included *E. coli* ATCC 25922, *Pseudomonas aeruginosa* ATCC 27853, and *Staphylococcus aureus* ATCC 25923.

The isolates were classified as resistant, intermediate resistant, or susceptible according to the zone sizes for all antibiotics except for polymyxins (HiMedia, Mumbai, Maharastra, India; Appendix A). Multidrug-resistant (MDR) bacteria are resistant to at least one agent in three or more antimicrobial categories, extensively drug-resistant (XDR) bacteria are resistant to at least one agent in all but two or fewer antimicrobial categories, and pan-resistant (PR) bacteria are resistant to all agents in all antimicrobial categories [17]. Resistant and intermediate-resistant strains were traditionally considered clinically resistant.

Selected bacteria were tested against additional antibiotics to fulfill the definition of XDR bacteria [17]. These included ceftaroline (anti-methicillin-resistant *Staphylococcus aureus* cephalosporin), cefazolin (non-extended-spectrum cephalosporin), and tigecycline (glycylcycline). Susceptibilities to ceftaroline (Allergan, Austin, TX, USA) and tigecycline (Sigma-Aldrich, St. Louis, MI, USA) (an MIC value of ≥0.5 mg/L for both) [18] were determined by the broth dilution method. Mueller–Hinton broth (Oxoid) was used as the medium dispensed in tubes. The starting concentration of the antibiotic used was 0.1 mg/L with increments of 0.1 mg/L in subsequent serial tubes. The bacterial inoculum was prepared as in the disc diffusion test. The tubes were incubated at 37 °C for 20 h before reading the results. The lowest concentration of the antibiotic that completely inhibited bacterial growth was taken as the MIC (minimum inhibitory concentration). Susceptibility to cefazolin (HiMedia) (a zone diameter of 20–22 mm for intermediate resistance and a zone diameter of ≤19 mm for resistance) [16] was performed by the disc diffusion method.

A Rapid Polymyxin NP test was conducted to determine the resistance to polymyxin B or polymyxin E (colistin). The test was performed according to the procedure of Nordmann et al. [19]. For negative control, ATCC 25922 *E. coli* was used, and for the positive control, a clinical isolate of *Morganella morganii* resistant to both polymyxin B and colistin was used. Resistance of test bacteria was indicated by growth in polymyxin-containing broth through a color change of a pH indicator.

Isolates having specific resistance patterns in susceptibility tests were further analyzed for specific genes encoding resistance by PCR assays (Appendix A). The dominant β-lactamase genes, *bla*_OXA-1_, *bla*_CMY-2_, *ampC*, *bla*_CTX-M_, *bla*_TEM_, and *bla*_SHV_ [20], were sought in isolates resistant to one or more of the β-lactams (cefepime, cephalothin, ceftazidime, cefoxitin, cefotaxime, ceftriaxone). If isolates were resistant to one or more of the carbapenems (ertapenem, meropenem, imipenem), they were screened for the dominant genes encoding carbapenem resistance (*bla*_NDM_, *bla*_OXA-48_, *bla*_KPC_, *bla*_IMP_, *bla*_VIM_, *bla*_GES_) [21,22].

Many PCR assays were performed to detect drug-resistant genes. Some procedures that were common to all PCR assays—preparation of template DNA, agarose gel electrophoresis, and detection of amplicons—are described below. A loopful of *E. coli* growth from MacConkey agar was boiled in PCR-grade sterile distilled water in an Eppendorf tube (Eppendorf, Hamburg, Germany) for 10 min. The supernatant containing the DNA was used as the template (~25 ng/μL as estimated via NanoDrop method (Thermo Scientific, Waltham, MA, USA). The following reagents were added in a PCR tube (Eppendorf) to obtain a 10 µL reaction volume: 2 µL of master mix (Solis BioDyne, Teaduspargi, Tartu, Estonia), 1 µL of forward primer (10 pmol), 1 µL of reverse primer (10 pmol), 5 µL of PCR-grade sterile distilled H_2_O, and 1 µL of diluted DNA (25 ng/µL). This PCR mix was then placed in a thermal cycler (Applied Biosystems, San Francisco, CA, USA) with the desired cycles. The primer sequences, cycling conditions, and amplicon sizes in PCR assays are shown in Appendix A. The amplicons were separated by agarose gel electrophoresis. The concentration of the agarose in the gel was 1.0% (for products ˃ 800 bp), 1.5% (for products 200–800 bp), and 2% (for products < 200 bp). The gel incorporated with ethidium bromide was run in Tris-Borate-EDTA buffer at a voltage of 100 V for ~40 min and photographed under UV light.

An ESBL NDP test was conducted, as described previously [23], for the expression of ESBL. This test is based on the hydrolysis of the β-lactam ring of cefotaxime, which generates a hydroxyl group, resulting in acidification of the medium with a change in color from red to yellow. The prevention of hydrolysis and, hence, the change in color of the medium by the addition of tazobactam indicates a positive test. A wild-type *E. coli* strain was used as the negative control, and a *Klebsiella pneumoniae* strain positive for CTX-M-15 was used as the positive control [24].

A Rapidec Carba NP test was used according to the manufacturer’s instructions (BioMerieux) for carbapenemase production. The substrate for carbapenemase was imipenem. The broth with imipenem also contained zinc to detect metallo-β-lactamase. Carbapenemase production will result in imipenem hydrolysis, resulting in color change in the medium due to a pH indicator. A broth devoid of imipenem was used as the negative control and a strain of *K. pneumoniae*-positive NDM-5 and OXA-244 was used as the positive control [25].

Global clones. Specific PCR assay was conducted on ExPEC isolates belonging to group B2 to detect ST131 clones [26]. ExPEC isolates belonging to group F were screened for three genes: *icd, gyrB,* and *uidA.* The ST648 clone was positive for the first two genes but negative for the third gene [27]. The primers for the assays are included in Appendix A.

Statistical tests. The difference in the proportions was tested via the chi-squared test or Fisher’s exact test. A *p*-value ≤ 0.05 was considered statistically significant.

## 3. Results

Antibiotic susceptibility. The data on antibiotic susceptibility are shown in Table 1. A high number (≥50%) of these isolates were resistant to cefepime, cephalothin, cefotaxime, streptomycin, amoxiclav, piperacillin, piperacillin/tazobactam, ampicillin, and tetracycline. The highest number of isolates was resistant to piperacillin, and the lowest number of isolates were resistant to fosfomycin. The high prevalence of resistance was influenced by the relatively high numbers of intermediate-resistance isolates.

The resistance of the isolates to the number and classes of antibiotics is shown in Table 2. Many isolates showed resistance to between 6 and 15 antibiotics. The number of cephalosporin-resistant *E. coli* isolates and carbapenem-resistant (CR) *E. coli* isolates were 136 (97.1%) and 14 (10.0%), respectively. Of the 28 antibiotics tested, one isolate each showed resistance to up to 20, 22, and 23 antibiotics. A total of 128 (91.4%) isolates were multidrug-resistant, being resistant to between 3 and 12 classes of antibiotics. One isolate each was resistant to 13 and 14 classes of antibiotics. These two isolates were resistant to additional antibiotics tested—ceftaroline, cefazolin, and tigecycline.

The resistance patterns of ExPEC, DEC, and global clones are shown in Appendix A. As the number of isolates in each category was small, a statistical comparison was not possible. A lack of resistance was found to one antibiotic (meropenem) in ExPEC; nine antibiotics (ertapenem, meropenem, imipenem, gentamicin, amikacin, fosfomycin, chloramphenicol, polymyxin B, and colistin) in DEC; and six antibiotics (meropenem, imipenem, gentamicin, fosfomycin, polymyxin B, and colistin) in global clones, ST131 and ST648. Both aEPEC isolates (J28 and H164) were resistant to four of the cephalosporins (cephalothin, ceftazidime, cefoxitin, and cefotaxime) and ampicillin. In addition, H164 was resistant to cefepime, ceftriaxone, aztreonam, streptomycin, all the penicillins with β-lactamase inhibitors, piperacillin, both antifolates, ciprofloxacin, azithromycin, and sulfafurazole. The ETEC isolate (H51) was susceptible to 25 of the 28 antibiotics tested. The isolate showed resistance to the other three antibiotics—cephalothin, cefotaxime, and tetracycline.

Differences in the resistance of carbapenem-resistant (CR) versus carbapenem-susceptible (CS) *E. coli* to other antibiotics are shown in Appendix A. In general, CR *E. coli* were more resistant than CS *E. coli.* This difference was significant for 10 antibiotics—cefepime, ampicillin, ampicillin/sulbactam, trimethoprim, co-trimoxazole, tetracycline, ciprofloxacin, fosfomycin, chloramphenicol, and colistin.

The prevalence of resistance in ExPEC and non-ExPEC (including nonpathogenic *E. coli* and DEC) to all except two antibiotics was similar with no significant differences. However, the higher prevalence of resistance in ExPEC for ceftriaxone and colistin was significant (Appendix A).

For PCR assays, six β-lactamase genes were targeted in a total of 136 samples that were resistant to ≥ one of the β-lactams (cefepime, cephalothin, ceftazidime, cefoxitin, cefotaxime, or ceftriaxone). All these samples were positive for *ampC* (100%). The next two frequent genes detected were *bla*_TEM_ and *bla*_CTX-M_, being present in 54 (39.7%) and 22 (16.2%) isolates, respectively. The least frequent genes were *bla*_OXA-1_, which was present in nine (6.6%) samples, and *bla*_CMY-2_, which was present in eight (5.9%) samples. None of the isolates were positive for the *bla*_SHV_ gene. One aEPEC (J28) and the ETEC (H51) isolates were positive for *ampC* only, while the other aEPEC (H164) had three genes: *ampC*, *bla*_CMY-2_, and *bla*_TEM_. For the 14 ExPEC that were all resistant to one or more of the cephalosporins, the following genes were present: *bla*_OXA-1_ in two (14.3%) isolates, *bla*_CMY-2_ in one (7.1%) isolate, *bla*_CTX-M_ in five (35.7%) isolates, *bla*_TEM_ in one (7.1%) isolate, and *ampC* in all fourteen (100.0%) isolates.

The ESBL NDP test showed that 18/136 (13.2%) isolates were ESBL-positive (Appendix A). All these isolates had no zone of inhibition in the disk diffusion test against the cefotaxime disk. All isolates were positive for *ampC*, 16 were positive for *bla*_CTX-M_, 6 were positive for *bla*_TEM_, 5 were positive for *bla*_OXA-1_, and 1 was positive for *bla*_CMY-2_. All were resistant to cephalothin, ceftazidime, cefotaxime, and cefepime. Sixteen isolates were additionally resistant to ceftriaxone and four to cefoxitin. None of the DECs were ESBL-positive, while 6 (42.9%) of the 14 ExPEC isolates were ESBL producers.

For the Rapidec Carba NP test, all 14 samples that were resistant to a carbapenem (the number of resistant strains will appear as 17 in Table 1 and Appendix A because one isolate was resistant to both ertapenem and meropenem, and another was resistant to all three meropenems) were tested, and all were negative.

Global clones. Since the ST131 clone belongs to the phylogenetic group B2 of ExPEC, all eight isolates of group B2 were tested, and one isolate (Z222) was positive. Since the clone ST648 belongs to group F of ExPEC, all five isolates of this group were tested, and three isolates (J211, H250, and J295) were positive. The resistance and ESBL production of these clones are shown in Appendix A.

## 4. Discussion

The water samples studied in different countries have been found to be contaminated with *E. coli* due to the disposal of improperly treated sewage [28].

We found that 91.4% of our *E. coli* isolates were multidrug-resistant and three isolates (2.1%) were resistant to ≥20 antibiotics. Further analysis showed that two isolates (1.4%) were resistant to 13–14 classes of antibiotics, in addition to ceftaroline, cefazolin, and tigecycline, which fulfills the definition of XDR bacteria [17]. The three additional antibiotics were not included in the initial screening of all isolates. Having found two isolates as possible candidates for extreme drug resistance, we tested these isolates against the additional antibiotics to fulfill the definition of extreme drug resistance. High resistance rates (>50%) to cephalosporins, in addition to ampicillin, piperacillin, and amoxiclav, were found. This correlates with the results of previous studies from Kuwait on clinical isolates [29,30,31,32]. Additionally, we found that most of the isolates were resistant to piperacillin/tazobactam, streptomycin, and tetracycline. It should be noted that isolates that exhibited intermediate resistance to most of the tested antibiotics are now considered as “susceptible, increased exposure category” when there is a high likelihood of therapeutic success if exposure to the antibiotic is increased by increasing the dosing regimen [33]. Inappropriate prescribing of antibiotics and self-medication are widespread risk factors in Kuwait [34]. These would have contributed to the high prevalence of resistance in *E. coli.*

All our *E. coli* isolates had the *ampC* gene. This gene is typically present in the chromosome with minimal constitutive enzyme production, although it can be hyper-expressed due to gene amplification or mutations creating a strong promoter. AmpC-type β-lactamases may also be carried on plasmids that mediate broad-spectrum cephalosporin resistance [35,36,37]. Further, 39.7% of *E. coli* isolates were *bla*_TEM_-positive, 16.2% had *bla*_CTX-M_, 6.6% and 5.9% were positive for *bla*_OXA-1_ and *bla*_CMY-2_, respectively, and none were *bla*_SHV_-positive. The ESBL producers in our study were 12.9%, which is similar to the finding of Dortet et al. [38]. Furthermore, 16 (88.9%) of 18 ESBL-positive isolates were *bla*_CTX-M_-positive, while *bla*_TEM_, *bla*_OXA-1_, and *bla*_CMY-2_ genes were present in 6 (33.3%), 5 (27.8%), and 1 (5.6%) isolates, respectively. All our isolates that were positive for the ESBL NDP test had no zone of inhibition to cefotaxime (in the disk diffusion test), the substrate used in the ESBL NDP test. It should also be noted that the *bla*_CTX-M_ gene, which is found in most of our ESBL producers, confers resistance to cefotaxime. This correlation between ESBL NDP test positivity, cefotaxime resistance, and *bla*_CTX-M_ positivity was also observed by Dortet et al. [38].

There are reports of CR *E. coli* causing human infections in Kuwait [39,40,41]. Fourteen (10.0%) of our *E. coli* isolates were CR, indicating a moderate prevalence of resistance to this class of antibiotics. CR strains were significantly more resistant to other antibiotics compared to CS strains. These findings correlate with those of other studies [42,43,44]. However, none of the CR *E. coli* isolates possessed carbapenemase genes, a finding like that of Zowawi et al. [45]. Performing PCR assays for selected genes may give false negative results as other known genes or novel genes could be missed. One way to confirm carbapenemase production is by performing a Rapidec Carba NP test that detects carbapenemase production regardless of specific genes. We determined that none of our 14 CR strains were positive in this test, thus confirming the absence of carbapenemase genes [46]. Other reported mechanisms of carbapenem resistance are efflux pumps [47] and loss of or reduction in the permeability of outer membrane proteins [48,49], which could be present in our isolates.

Among the DEC, the two aEPECs were multidrug-resistant and possessed cephalosporin resistance genes. The lone ETEC was comparatively more susceptible. None of the DEC isolates was an ESBL producer. DEC isolated from sewage treatment plants or environmental water samples from other countries showed resistance to many antibiotics [50,51,52,53].

It was noted that there were no significant differences between ExPEC isolates and non-ExPEC isolates in the resistance rates except for resistance to two antibiotics. ExPEC cultured from sewage effluent or river water receiving sewage in South Africa or Mexico showed different patterns of resistance to antibiotics [50,51,52,53].

The ST131 lineage has more virulence genes compared to other *E. coli* strains and exhibits diverse mechanisms of antibiotic resistance [54,55,56,57]. It has been found in clinical samples and in wastewater and recreational water samples [58]. This clone carries the β-lactamase gene, *bla*_CTX-M_ [59]. In addition, some strains might carry *bla*_TEM_ [60,61], *bla*_SHV_ [61,62], *ampC* [60], or other genes [60]. In the strains of this clone, CTX-M is usually coproduced with OXA-1 and an aminoglycoside-modifying enzyme encoded by the *aac(6*′)-*Ib*-*cr* gene that confers resistance to aminoglycosides and fluoroquinolones [55,63]. CTX-M was shown to enhance the survival of the bacterium [13,64]. ST131 was previously reported in Kuwait in clinical samples [65].

The other global clone, ST648, is known to harbor resistance genes such as *bla*_CTX-M_, *bla*_CMY-2_, *bla*_OXA-48_, and *bla*_NDM_ [66]. This ST has also been isolated from fresh and environmental water samples [67,68,69] and from wastewater samples [70,71,72].

A total of 4 out of 14 of our ExPEC isolates (28.6%) were of ST131 and ST648 lineages. They were resistant to 6–13 classes of antibiotics and possessed resistant genes, and two were ESBL producers. All four of these ExPEC isolates had a combination of three ExPEC genes [14], suggesting a high virulence in these clones, as found by other authors [54,55,56,57].

Thus, the reuse of water from treated sewage and the discharge of untreated sewage into the seas pose a threat to human health in Kuwait via the finding of pathogenic (DEC and ExPEC) and resistant *E. coli* in raw sewage. The characteristics of these bacteria appear to reflect those of the isolates from human infections in the community. To our knowledge, such a comprehensive study of sewage *E. coli* for drug resistance to a wide range of antimicrobial agents, including resistance mechanisms to extended-spectrum beta-lactams and carbapenems, and global clones has not been conducted before in Kuwait or in the region. The finding of MDR and XDR *E. coli* isolates constitutes a threat to public health as they can cause infections that are difficult to treat. A limitation of our study is that we have not thoroughly investigated the mechanisms of resistance. Leading experts have declared that antimicrobial resistance is a global health emergency [73]. The spread of drug-resistant bacteria in the community increases the risk of death for common infections, such as urinary tract infections, for which *E. coli* is the predominant pathogen [74]. Emerging technologies, such as network pharmacology and functional genomics, in association with in vivo imaging platforms, offer great promise for new antimicrobial discovery [75]. The recent discovery of some antibiotics—fabimycin [76] and teixobactin [77]—offers new hope for treating multi-resistant superbugs. Our findings also have relevance to one health. One health is an approach that recognizes that the health of people is closely connected to the health of animals and our shared environment [78]. The microorganisms in the sewage are derived from humans, animals, plants, and soil, and have an impact on these constituents. Thus, the presence of antibiotic-resistant *E*. *coli* in sewage in Kuwait has implications for one health.

## 5. Conclusions

The analysis of sewage in Kuwait has shown the presence of MDR and XDR *E. coli* and virulent *E. coli* global clones in sewage. Since some raw sewage is discharged into coastal waters that are used for recreational activities, the presence of MDR and XDR *E. coli* and global clones in the coastal waters poses a danger to human health, which may result in difficult-to-treat infections.

## Figures and Tables

**Table 1 microorganisms-11-02610-t001:** Susceptibility and resistance of 140 *E. coli* isolates to various antimicrobial agents.

Antimicrobial Agents	No. of Isolates with Indicated Resistance		
	Resistant (R)	Intermediate Resistance (IR) ^a^	Susceptible (S)
Cefepime	24	46	70
Cephalothin	92	39	9
Ceftazidime	29	38	73
Cefoxitin	11	19	110
Cefotaxime	77	42	21
Ceftriaxone	27	15	98
Aztreonam	24	17	99
Ertapenem	3	1	136
Meropenem	2	0	138
Imipenem	11	0	129
Gentamicin	5	2	133
Streptomycin	48	30	62
Amikacin	6	14	120
Amoxiclav	63	18	59
Ampicillin/Sulbactam	26	24	90
Piperacillin/Tazobactam	20	55	65
Ampicillin	85	22	33
Piperacillin	88	30	22
Trimethoprim	59	1	80
Co-Trimoxazole	57	1	82
Tetracycline	66	15	59
Ciprofloxacin	22	24	94
Fosfomycin	2	2	136
Azithromycin	13	0	127
Chloramphenicol	21	6	113
Sulfafurazole	58	8	74
Polymyxin B ^b^	25	NA	115
Colistin ^b^	4	NA	136

^a^ In the counting of resistant strains, R and IR strains were merged. ^b^ Susceptibility to polymyxin B and colistin was measured by rapid polymyxin NP test for which the IR category is not applicable (NA).

**Table 2 microorganisms-11-02610-t002:** Resistance ^a^ of 140 *E. coli* isolates to number of antibiotics and their classes.

Resistance to Antibiotic and Antibiotic Class	No. of Isolates
Resistance to no. of Antibiotic(s)	No. of Isolates
0	2
1	2
2	2
3	8
4	6
5	8
6	10
7	8
8	8
9	8
10	7
11	8
12	12
13	12
14	10
15	12
16	5
17	5
18	3
19	1
20	1
22	1
23	1
Resistance to no. of antibiotic class(s)	No. of isolates
0	2
1	3
2	5
3	12
4	13
5	15
6	15
7	13
8	17
9	18
10	15
11	7
12	3
13	1
14	1

^a^ Include both resistant and intermediate-resistance isolates.

## Data Availability

The data presented in this study are available on request from the corresponding author.

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
