# Peer review of "Multidrug-Resistant and Extensively Drug-Resistant Escherichia coli in Sewage in Kuwait: Their Implications"

_microorganisms, 2023, doi:10.3390/microorganisms11102610_

Round 1

Reviewer 1 Report

Authors here present an interesting study on muli-drug resistant E. coli. The experimental deisgn and presentation is good but lacks comparative studies with other bacteria and in different physiological conditions like, aerobic and anaerobic, temperature and pH as these conditions vary in human body depending on the site they reside and infect.

It is stange to see there are no figures in whole MS. can authors provide some ?

English is good.

Author Response

Comments.

Authors here present an interesting study on multi-drug resistant E. coli. The experimental design and presentation is good but lacks comparative studies with other bacteria and in different physiological conditions like, aerobic and anaerobic, temperature and pH as these conditions vary in human body depending on the site they reside and infect.

Response. The focus of our study is E. coli whose habitat is the large intestine. What the referee suggests is broad and beyond the scope of our study.

It is strange to see there are no figures in whole MS. can authors provide some?

Response. Our data involve comparison of numbers with statistics. Tabulating the data gives accurate information for the readers to evaluate the data. From figures, readers can infer approximations only. Hence the reason for giving all data in tables.

We have examined the text again, and made minor editorial corrections as suggested by the referee.

We thank the referee for the constructive criticisms.

Reviewer 2 Report

The manuscript presented by Redha and colleagues presents the results of the analysis of 140 isolates of E. coli obtained from sewage samples. The data presented reflects how the panorama of multidrug resistance to antibiotics is at a critical stage in out-of-hospital environments. Overall, the manuscript is well written and the results are well presented. Below are some considerations.

Results

Lines 151-154: I believe there is a mistake in mentioning that more than 50% of the isolates were resistant to cefepime, streptomycin, amoxicillin-clavulanate, piperacillin-tazobactam, and tetracycline. From my point of view, the authors are adding up the percentage of resistance with that of intermediate resistance to these antibiotics. This is not a recommended practice, considering that there has been a reclassification from "intermediate resistant" to "sensitive by increasing exposure", even if the name has been retained. I suggest you review it to avoid wrong conclusions.

Discussion

In the last paragraph of the discussion, it would be interesting to include the significance of the results in the context of One Health.

Author Response

Comments.

Results

Lines 151-154: I believe there is a mistake in mentioning that more than 50% of the isolates were resistant to cefepime, streptomycin, amoxicillin-clavulanate, piperacillin-tazobactam, and tetracycline. From my point of view, the authors are adding up the percentage of resistance with that of intermediate resistance to these antibiotics. This is not a recommended practice, considering that there has been a reclassification from "intermediate resistant" to "sensitive by increasing exposure", even if the name has been retained. I suggest you review it to avoid wrong conclusions.

Response. This sentence has been added in the Results. " The high prevalence of resistance was influenced by the relatively high numbers of intermediate resistance isolates" (L181-183 in tracked changes manuscript)).

In the Discussion, the following sentence has been added. "It should be noted that intermediate resistant isolates to most of the tested antibiotics are now considered as "susceptible, increased exposure category" when there is a high likelihood of therapeutic success if exposure to the antibiotic is increased by increasing the dosing regimen (33)" (L269-272 in tracked changes manuscript)

Discussion

In the last paragraph of the discussion, it would be interesting to include the significance of the results in the context of One Health.

Response. The following text has been added. "Our findings also have relevance to one health. One health is an approx. that recognizes that the health of people is closely connected to the health of animals and our shared environment [78]. The microorganisms in the sewage are derived from humans, animals, plants, and soil, and have impact on these constituents. Thus, the presence of antibiotic-resistant E. coli in sewage in Kuwait has implications for one health" (L352-357 in tracked changes manuscript).

We are grateful to the referee for the constructive suggestions.